# Role of Intestinal Inflammation and Permeability in Patients with Acute Heart Failure

**DOI:** 10.3390/medicina60010008

**Published:** 2023-12-20

**Authors:** Marcello Covino, Antonella Gallo, Noemi Macerola, Erika Pero, Francesca Ibba, Sara Camilli, Laura Riccardi, Francesca Sarlo, Grazia De Ninno, Silvia Baroni, Francesco Landi, Massimo Montalto

**Affiliations:** 1Department of Emergency Medicine, Fondazione Policlinico Universitario “A. Gemelli”, IRCCS, Università Cattolica del Sacro Cuore, Largo A. Gemelli, 8, 00168 Rome, Italy; marcello.covino@policlinicogemelli.it; 2Department of Geriatrics and Orthopedics, Fondazione Policlinico Universitario “A. Gemelli”, IRCCS, Largo A. Gemelli, 8, 00168 Rome, Italy; pero.erika@gmail.com (E.P.); franci.ibba@gmail.com (F.I.); sara.camilli05@gmail.com (S.C.); francesco.landi@unicatt.it (F.L.); massimo.montalto@unicatt.it (M.M.); 3Division of Internal Medicine, San Carlo di Nancy Hospital, GVM Care and Research, Via Aurelia, 275, 00165 Rome, Italy; noemi.macerola@gmail.com; 4Department of Medical and Surgical Sciences, Fondazione Policlinico Universitario “A. Gemelli”, IRCCS, Largo A. Gemelli, 8, 00168 Rome, Italy; laura.riccardi@policlinicogemelli.it; 5Department of Chemistry, Biochemistry and Clinical Molecular Biology, Fondazione Policlinico Universitario “A. Gemelli”, IRCCS, Largo A. Gemelli, 8, 00168 Rome, Italy; francesca.sarlo@guest.policlinicogemelli.it (F.S.); silvia.baroni@policlinicogemelli.it (S.B.); 6Department of Chemistry, Biochemistry and Clinical Molecular Biology, Università Cattolica del Sacro Cuore, Largo A. Gemelli, 8, 00168 Rome, Italy; grazia.deni@gmail.com; 7Faculty of Medicine, Università Cattolica del Sacro Cuore, Largo A. Gemelli, 8, 00168 Rome, Italy

**Keywords:** heart failure, intestinal inflammation, permeability, fecal calprotectin, zonulin, inflammaging

## Abstract

*Background and Objectives*: Heart failure (HF) represents a major health burden. Although several treatment regimens are available, their effectiveness is often unsatisfactory. Growing evidence suggests a pivotal role of the gut in HF. Our study evaluated the prognostic role of intestinal inflammation and permeability in older patients with acute HF (AHF), and their correlation with the common parameters traditionally used in the diagnostic-therapeutic management of HF. *Materials and Methods*: In a single-center observational, prospective, longitudinal study, we enrolled 59 patients admitted to the Emergency Department (ED) and then hospitalized with a diagnosis of AHF, from April 2022 to April 2023. Serum routine laboratory parameters and transthoracic echocardiogram were assayed within the first 48 h of ED admission. Fecal calprotectin (FC) and both serum and fecal levels of zonulin were measured, respectively, as markers of intestinal inflammation and intestinal permeability. The combined clinical outcome included rehospitalizations for AHF and/or death within 90 days. *Results*: Patients with increased FC values (>50 µg/g) showed significantly worse clinical outcomes (*p* < 0.001) and higher median levels of NT-proBNP (*p* < 0.05). No significant correlation was found between the values of fecal and serum zonulin and the clinical outcome. Median values of TAPSE were lower in those patients with higher values of fecal calprotectin (*p* < 0.05). After multivariate analysis, NT-proBNP and FC values > 50 µg/g resulted as independent predictors of a worse clinical outcome. *Conclusions:* Our preliminary finding supports the hypothesis of a close relationship between the gut and heart, recognizing in a specific marker of intestinal inflammation such as FC, an independent predictive prognostic role in patients admitted for AHF. Further studies are needed to confirm these results, as well as investigate the reliability of new strategies targeted at modulation of the intestinal inflammatory response, and which are able to significantly impact the course of diseases, mainly in older and frail patients.

## 1. Introduction

Heart failure (HF) represents one of the leading causes of mortality and morbidity worldwide [1]. In particular, acute HF (AHF) is the first cause of unplanned hospitalization in people >65 years [2,3]. In the current European Society of Cardiology (ESC) HF guidelines, AHF is defined as a rapid or gradual onset of symptoms and/or signs of HF, severe enough for the patient to seek urgent medical attention, leading to an unplanned hospital admission or an emergency department (ED) visit [1].

Despite the significant efforts in improving the management of these patients, however, the results remain far from being satisfactory.

Systemic inflammation has been recognized as a common pathological feature of both AHF and chronic HF (CHF) [4]. It has been associated with disease development, progression, and complication, and represents an independent predictive parameter of poor prognosis [4]. These findings have inevitably raised the question of targeting inflammation for therapeutic purposes in HF. In the context of both AHF and CHF, growing evidence has reported increased serum levels of inflammatory markers, such as C-reactive protein (CRP), interleukins 1 and 6 (IL-1, IL-6), and tumor necrosis factor-alpha (TNF-α), as well as elevated lipopolysaccharide (LPS) circulatory levels [5,6,7], suggesting the presence of a persistent “low-grade inflammation” [8]. In particular, AHF seems to be characterized by the upregulation of specific inflammatory pathways leading to worse cardiac structure and function, as well as worse prognosis, more pronounced than in CHF [7,8,9].

Recently, other proteins have received the attention of researchers, such as calprotectin (also known as S100A8/A9, MRP-8/14 protein, calgranulin A/B), a protein widely represented in the cytoplasm of neutrophils and monocytes and released into the extracellular space after pro-inflammatory stimuli [10,11]. Serum calprotectin levels have been correlated with severe CHF (New York Heart Association, NYHA class III-IV), mainly in patients with ischemic etiology [12].

Since Tang’s first statement of the “gut hypothesis of heart failure” [13], growing evidence suggests a pathophysiological pivotal role of the gut in HF [5,14,15,16]. In particular, both in CHF, and especially in AHF, venous congestion and reduced cardiac output induce bowel hypoperfusion, leading to mucosal ischemia and hypoxic injury to the intestinal villous structures. The consequent impairment in intestinal barrier function increases bowel wall permeability, leading to bacterial translocation into the circulatory system and increased levels of circulating endotoxins, such as LPS, thus promoting a typical inflammatory state [17].

However, no data are available about the specific markers of intestinal inflammation and permeability in the context of both AHF and CHF. Fecal calprotectin represents a specific and widely recognized marker of intestinal neutrophilic inflammation, closely correlated with the histological grade of bowel inflammation [10]. Both the high predictive negative value and reliability in specifically assessing the “subclinical intestinal inflammation” make this fecal marker a valid and very useful tool in the management of bowel inflammatory diseases [10,11].

On the other hand, gut barrier integrity is normally modulated by several factors such as tight junctions and regulatory proteins, such as zonulin [18]. Up to now, serum and fecal zonulin levels have been assessed as a surrogate marker of intestinal permeability, involved in the pathogenesis of intestinal and extraintestinal pathological conditions [19,20,21,22,23,24]. To our knowledge, only one study evaluated serum zonulin in HF, without showing a significant correlation with the clinical stage of the disease [25,26]. To date, no data are reported on fecal zonulin in the context of AHF or CHF.

In this study, we evaluated the prognostic role of intestinal inflammation and permeability in older patients with AHF, and their correlation with the common parameters traditionally used in the diagnostic-therapeutic management of HF.

## 2. Patients and Methods

This is a single-center observational, prospective, longitudinal pilot study, of patients attending the Emergency Department (ED) of a tertiary urban University Hospital over a period from April 2022 to April 2023, and then hospitalized. We included consecutive patients who received a definitive diagnosis of AHF formulated by the ED Clinicians, according to current guidelines [1] and to the ED Cardiology Consultant’s evaluation. AHF was defined as “de novo” or acute decompensation of CHF.

Exclusion criteria included: minor age (age < 18 years), history of primary hepatopathy (steatosis or steatohepatitis, viral hepatitis, cirrhosis, hepatocellular carcinoma, or history of alcohol abuse), history of chronic inflammatory bowel disease (IBD), acute diverticulitis or infective/ischemic colitis, active celiac disease, chronic kidney disease undergoing dialysis treatment, history of congenital heart malformations, or patients undergoing corrective cardiac surgery.

All the enrolled patients underwent a series of clinical, laboratory, and echocardiographic evaluations within the first 48 h of ED admission (baseline) (Figure 1). The clinical severity of AHF was assessed according to the NYHA classification [1].

The following laboratory parameters were assayed within the first 48 h of ED admission: the N-terminal fragment B-type natriuretic peptide (NT-proBNP), CRP, IL-6, alanine aminotransferase (ALT), gamma-glutamyl transferase (γGT), alkaline phosphatase (ALP), and total bilirubin. Fecal calprotectin was measured as a marker of intestinal inflammation; serum and fecal levels of zonulin were also measured as markers of intestinal permeability.

All patients underwent transthoracic echocardiogram for the evaluation of left ventricular systolic function and ejection fraction (EF), right ventricular function via tricuspid annular plane excursion (TAPSE), systolic pulmonary artery pressure (PAPs), and inferior vena cava (IVC) diameter. The presence of ascites was also recorded.

According to the current guidelines, the total cohort was divided into four subgroups based on the main clinical presentation, i.e., patients with acute decompensated heart failure (ADHF), acute pulmonary edema (OPE), isolated right ventricular failure (IRVF), and cardiogenic shock (CS) [1,2,3,27,28].

Finally, at 30, 60, and 90 days after ED admission, patients were re-evaluated via phone interview, to record the occurrence of rehospitalizations for AHF and/or death from cardiovascular events.

### 2.1. Outcome Measures

Our study evaluated the prognostic role of intestinal inflammation and permeability in patients with AHF.

As a secondary endpoint, we evaluated the correlation of the intestinal inflammatory marker with the other laboratory and instrumental parameters traditionally used in the diagnostic-therapeutic management of AHF.

### 2.2. Fecal Calprotectin Test

Stool samples were collected in suitable containers and frozen at −80 °C until the assay. They were then thawed and prepared using the LIAISON^®^ Q.S.E.T. Device Plus (REF 319060) containing a specific extraction buffer for the calprotectin assay, and then shaken for 40 min. The analysis was performed on a LIAISON^®^ XL Analyzer using DiaSorin (Saluggia, Italy) LIAISON^®^ Calprotectin in a chemiluminescence (CLIA) kit, sandwich-type, employing 2 monoclonal antibodies for the quantitative determination of calprotectin. In this system, the light signal is measured by a photomultiplier in relative light units (RLU) and is proportional to the concentration of calprotectin present in the samples based on the values obtained from the calibration curve. The measurement range of the method is 5–800 µg/g and the normality cut-off is 50 µg/g. The intra-assay and inter-assay CV are <8%.

### 2.3. Serum and Fecal Zonulin Test

Serum samples were obtained after centrifugation at 4000 rpm for 10 min, aliquoted, and frozen at −80 °C; before analysis, they were brought to room temperature and vortexed. The stool samples, collected in suitable containers, were prepared in a 15 mL falcon by adding 1 g of fecal material to 9 mL of PBS buffer (phosphate buffered saline, pH 7.4), according to the directions provided by the Human Zonulin ELISA kit (BT LAB-Bioassay Technology Laboratories, Shanghai, China), specifically for the assay of zonulin on serum and stool. The intra-assay and inter-assay CV are <10%.

The measurement range of the method is 0.13–90 ng/mL and the normality cut-off for serum is <48 ng/mL and for stool is <107 ng/g. For stool samples, the zonulin concentration was obtained by multiplying the result in ng/mL (as reported by the kit) by 2.07.

### 2.4. Statistical Analysis

Continuous variables are reported as medians (interquartile range, IQR), while categorical variables are expressed as absolute numerical values (%). Univariate analysis was performed using Mann–Whitney’s U test for continuous variables, and the chi-squared test for categorical variables. A value of *p* < 0.05 was considered statistically significant.

The parameters significantly associated with the study endpoint at univariate analysis were entered into a multivariate logistic regression model. In light of the reduced number of cases in the analysis, the continuous parameters were dichotomized to improve the regression model fitting and parameter estimation. Since all the patients in the study group had NT-proBNP values above the standard reference value, we dichotomized the cases using a cut-off of 4874 pg/mL chosen by the Youden index J of a Receiver Operating Characteristics analysis performed based on the association between NT-proBNP and death/readmission within 90 days (Appendix A). For fecal calprotectin values, the patients were dichotomized using the reference value (50 µg/g). To evaluate the coherence in the study design, and to prove the goodness of our model, we also performed a separate logistic regression analysis dichotomizing the calprotectin value via ROC analysis (Appendix A and Appendix A).

The predictors exceeding the 5 thresholds for variance inflation due to multicollinearity were excluded from the model, to reduce the degrees of liberty of the final logistic models to obtain a more reliable parameter estimation. Since NT-proBNP showed high collinearity with the NYHA class and TAPSE value, only the former variable was entered into the models. Since two variables were entered into the final logistic models, the number of events in the sample size (19 events) was sufficient for the parameter estimation.

All statistical analyses were performed using the 64-bit version SPSS 25.0 (IBM, New York, NY, USA) and MedCalc v18 (MedCalc Software Ltd., Ostend, Belgium).

### 2.5. Ethical Statement

The study protocol was approved by the Ethics Committee of the Fondazione Policlinico Universitario A. Gemelli IRCCS (Prot. ID 4819/2022). Each patient signed an informed consent before joining the study.

## 3. Results

### 3.1. Sample Study

During the study period, we enrolled 75 subjects admitted to the ED for AHF and then hospitalized in the Medical Wards of Fondazione Policlinico Agostino Gemelli IRCCS. However, 16 subjects were excluded from the final analysis. Among them, eleven patients gave their informed consent at the time of admission to the ED, but were unable to provide stool samples within the pre-established 48 h. Instead, five patients did not complete the total 90-day follow-up by phone interview. Three of them were patients discharged from hospital to a nursing home, while two patients who were discharged home were unresponsive to phone calls. Thus, our final study cohort included 59 subjects (23 Females, 36 Males) with a median age of 81 (76–86) years. All demographic, clinical, laboratory, and transthoracic echocardiographic parameter data are reported in Table 1.

At the basal evaluation, median values of fecal zonulin were 54.4 ng/g (42.0–78.4) and of serum zonulin were 8.3 ng/mL (7.0–9.6), both of which resulted in the normal range (i.e., <107 ng/g for stool and <48 ng/mL for serum samples). Conversely, median values of fecal calprotectin were 117 µg/g (47–170), above the normal cut-off represented by 50 µg/g. In particular, 36 patients (61%) showed increased fecal calprotectin levels, while 23 patients (39%) showed normal values. Table 1 shows the correlation between fecal calprotectin values (normal or increased) and the other clinical and echocardiographic parameters. In particular, patients with higher fecal calprotectin levels were older (*p* < 0.05) and showed significantly higher median levels of NT-proBNP (*p* < 0.05). Moreover, median values of TAPSE were higher in those patients with normal values of fecal calprotectin (*p* < 0.05).

No other significant differences were reported in the remaining analysis, as shown in Table 1.

The number of patients with a reduced ejection fraction (EF < 40%) was 42 (71.2%). None of the enrolled patients presented “de novo” HF. According to the current guidelines [1], most of the subjects showed a clinical scenario of ADHF (48 patients, 81.4%). Eight patients (8.5%) showed OPE, two patients (3.4%) had isolated RV failure, and one patient (1.7%) had CS (Table 1). In consideration of the most frequent phenotype encountered, the mainstay of treatment consisted of early intravenous (i.v.) loop diuretics and oxygen supplementation. Four of the eight patients with OPE also needed early i.v. vasodilators (nitroglycerin) and non-invasive positive-pressure ventilation. Only one patient received inotropes and vasopressors (Table 2).

Also, HF medications at discharge are reported in Table 2.

### 3.2. Predictors of Clinical Outcome

Out of 59 patients who completed the follow-up, 19 (32%) died or were readmitted for AHF within the following 90 days since the baseline. In particular, eight patients died (13.5%), seven of them within the first 30 days and one of them between 61 and 90 days. A total of 11 patients (18.6%) were readmitted within the 90 days since the baseline. In particular, seven patients underwent single rehospitalization (one within the first 30 days, three between 31 and 60 days, and three between 61 and 90 days) and four patients underwent multiple rehospitalizations during the follow-up. So, the total number of rehospitalizations was 16 (Figure 2).

Table 3 shows the correlation among all the studied parameters and the composite clinical outcome. By using univariate analysis, a significant correlation was found for median basal values of NT-proBNP (*p* < 0.001). In addition, clinical stratification by NYHA classification showed that patients belonging to the NYHA classes <IV showed better outcomes compared to those patients belonging to the NYHA class IV (*p* < 0.001).

Regarding the parameters of intestinal inflammation and permeability, a significant correlation was found between fecal calprotectin and the clinical combined outcome. In particular, when patients were divided according to normal (<50 µg/g) or increased (>50 µg/g) values of fecal calprotectin, the subgroup of patients with the higher values of this fecal marker showed significantly worse clinical outcomes compared to the other ones (*p* < 0.001, Table 3 and Figure 2).

Otherwise, serum and fecal zonulin values all resulted within normal ranges, both in those patients who died or were readmitted to hospital within 90 days, and in those patients for whom neither was the case. Therefore, no significant correlation was found among the values of fecal and serum zonulin, and the clinical outcome (Table 2).

Multivariate analysis by logistic regression confirmed the prognostic independent predictive role of fecal calprotectin values over 50 µg/g and median NT-proBNP values (Table 4).

## 4. Discussion

Our study shows that fecal calprotectin, a well-known marker of clinical and subclinical intestinal inflammation, could represent an independent prognostic marker in patients admitted for AHF.

To date, only a few studies are available regarding the role of calprotectin in HF [12,29]. Serum calprotectin has been recognized as a valid biomarker in predicting one-year mortality, playing a significant prognostic role mainly when used in older populations. In particular, Ma et al. found in 54 older patients (median age 82 years) with CHF belonging to NYHA classes III-IV, that the combination of S100A8/A9 (i.e., calprotectin) and IL-6 levels increased the one-year and six-month mortality predictive strength of BNP, IL-8, TNF-α [29].

The exact mechanisms underlying the role of calprotectin in HF remain unclear. It has been suggested that the serum complex, S100A8/A9, may stimulate cytokine release, amplifying the inflammatory response by the NF-kB e p38 MAPK pathway activations [30]; consistently, various genetic and pharmacological strategies aimed at the disruption of the S100A8/A9-Nlrp3-IL-1β signaling axis have been shown to dampen myelopoiesis and improve cardiac function after ischemic damage [31].

On the other hand, there are no data about the role of fecal calprotectin in the context of both AHF and CHF. Conversely to serum calprotectin, fecal calprotectin is widely used in clinical practice as an expression of a specific inflammatory process at the intestinal level, potentially representing a target for therapeutic interventions. It is well-known that in HF patients, in particular in the acute phase, gut luminal hypoxia and changes in local pH induced by intestinal hypoperfusion may significantly contribute to changes in the gut microbiota composition with a decrease in bacteria with anti-inflammatory functions [14,15]. Interestingly, the increase in pathogenic bacteria was correlated with a worsening NYHA class [32]. However, neither the real composition of microbiota in patients with HF, nor changes occurring between AHF and CHF are completely clear; despite encouraging results, therapeutic strategies based on microbiota modulation are still far from being validated in clinical practice [33,34].

We found that the median values of fecal calprotectin resulted above the normal range in the total cohort of AHF patients. As we excluded patients with a history of bowel diseases, it is plausible to hypothesize that most of our AHF patients showed a persistent grade of systemic “low-grade inflammation”, which was also reflected at the intestinal level. Whether this element plays a pathogenetic role in the progression of disease may be worthy of further investigation. In any case, according to our preliminary results, higher fecal calprotectin values at the baseline may influence the course of the disease and predict a worse prognosis.

Increased intestinal permeability has also been closely correlated with the clinical severity of HF. In this regard, Pasini et al. showed that in 60 patients with mild CHF (NYHA functional class I to II) and moderate to severe CHF (NYHA functional class III to IV) that, compared with normal control subjects, the entire CHF population showed significant increases in intestinal permeability (assessed by cellobiose sugar test) and venous blood congestion (expressed as by echocardiography), together with systemic inflammation parameters (CRP). Interestingly, these differences were more pronounced in patients with moderate to severe NYHA functional classes than in patients with mild NYHA functional classes [14]. In our work, we found values of both serum and fecal zonulin within the normal range. However, conversely to fecal calprotectin, the reliability of zonulin assessment as a surrogate marker of intestinal permeability is still controversial and mainly based on studies focused on intestinal conditions, such as active celiac disease, clearly characterized by increased gut permeability. However, data are not conclusive about the dosage of both serum and fecal zonulin in patients with inflammatory bowel diseases [21]. Despite these uncertainties, we chose to measure zonulin levels instead of other validated methods for the intestinal permeability assessment (such as the 51Cr-EDTA scintigraphy or the lactulose/mannitol ratio), in light of their unfeasibility in the context of an ED.

In the last few years, AHF classification was mainly based on the congestion–hypoperfusion phenotypes (“wet and dry”, “cold and warm”), with the “wet and warm” group accounting for nearly 80% of cases and the “cold and dry” group for less than <1% [1,2,3,27,28]. This classification has been partially modified in the current guidelines [1], via the identification of the three main mechanisms involved, i.e., systemic congestion, pulmonary congestion, and tissue hypoperfusion, and the consequent clinical scenarios without significant overlapping [1,2,3,27,28]. Early identification of the AHF clinical profile represents the cornerstone in the management of HF in order to assess the best hemodynamic-based medical strategies [1,2,3,27,28]. The BNP and NT-proBNP have a well-recognized prognostic role, both in terms of mortality and rehospitalization due to cardiac causes [35,36], as also confirmed in our population. Natriuretic peptides are recommended to be measured in all patients admitted for AHF, and conversely, other blood tests and markers should be used, such as procalcitonin, D-dimer, and lactate, that would be recommended based on the clinical scenario [1,2,3,27,28].

However, the role of natriuretic peptides to guide treatment remains unclear [36], as their values are influenced by many factors such as age, kidney function, and obesity. Therefore, multi-parametric strategies, reflecting different HF pathways, have been encouraged to better characterize the disease phenotypes of each individual patient, and consequently, to modulate the therapeutic approach [36]. Up to now, only a few biomarkers have shown an additive prognostic value to NPs, such as high-sensitivity troponins and the soluble suppression of tumorigenesis-2 [36]. Little, if none, is known about the role in this topic of fecal calprotectin, which represents more specifically a marker of intestinal inflammation rather than congestion. In our study, we found that both NT-proBNP and fecal calprotectin values were independent prognostic markers, and that they were intrinsically related to each other. It is well-known that an optimal management of AHF patients should be tailored according to the clinical presentation and congestion–hypoperfusion phenotypes. However, clinical overlapping is not so uncommon, so the more parameters we have to characterize our AHF patients’ profiles, the more we may be able to identify targeted, and hopefully, efficacious strategies. It would be interesting to verify in larger studies whether subjects with higher levels of fecal calprotectin may benefit from i.v. diuretics rather than oral administration, as the expression of impaired oral drug absorption is not only related to systemic and local congestion but also to a specific inflammatory process occurring at the intestinal level.

Our cohort mainly included older subjects, with a median age of 81 years. Although we do not have a younger control population, it is plausible that the presence of subclinical intestinal inflammation can be incorporated as part of the process of age-related chronic low-grade inflammation, the so-called “inflammaging”. Chronic inflammation is a widely recognized risk factor across cardiovascular disease, and, mainly in older subjects, it can be associated with a greater loss of muscle mass and strength, contributing to the onset of frailty and sarcopenia and the consequent relevant impact on morbidity and mortality [37].

Finally, we found that patients with higher levels of fecal calprotectin also showed lower and under the normal range of TAPSE values, suggesting that splanchnic congestion derived from right ventricular failure may contribute to triggering an inflammatory intestinal process, as revealed by fecal calprotectin measurement.

Our study has different limitations. First of all, due to the small sample size and the heterogeneity of cardiac disease etiology, our work should be considered as a pilot study, and larger studies are needed to confirm our results, and to identify eventual significant differences of fecal calprotectin among the main AHF phenotypes. Secondly, we did not collect any data on the microbiota composition; therefore, we were not able to stratify our cohort according to specific microbiota phenotypes for a targeted approach. Third, we did not make a longitudinal assessment of laboratory and intestinal parameters, so we could not evaluate whether the persistence of low-grade intestinal inflammation during the follow-up may influence the prognosis and we have no data about the eventual change in discharge therapy. Finally, we do not have a control group of younger subjects with AHF, although there is still no evidence about significant modifications of FC values depending on the different age groups.

## 5. Conclusions

In conclusion, our preliminary findings support the hypothesis of a close relationship between the gut and heart, recognizing in a specific marker of intestinal inflammation such as fecal calprotectin, an independent predictive prognostic role in patients admitted for AHF.

Further and larger studies are needed to confirm these results, the eventual role of fecal calprotectin as a guide to HF therapy, and the eventual differences according to age group. Far from identifying new successful strategies, these findings may encourage research on targeted inflammatory response modulation at the intestinal level, perhaps positively impacting the course of the disease, mainly in older and frailer patients.

## Figures and Tables

**Figure 1 medicina-60-00008-f001:**
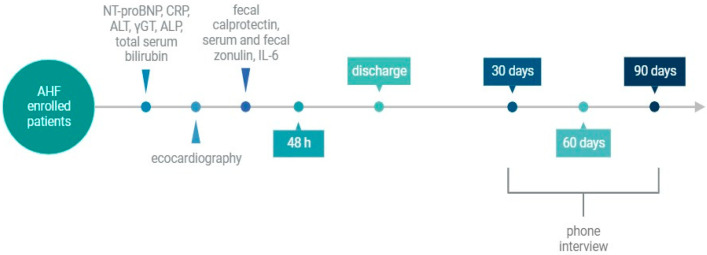
Design of the study. Abbreviations: NT-proBNP: N-terminal fragment B-type natriuretic peptide, CRP: C-reactive protein, ALT: alanine aminotransferase, γGT: gamma-glutamyl transferase, ALP: alkaline phosphatase, IL-6: Interleukin-6.

**Figure 2 medicina-60-00008-f002:**
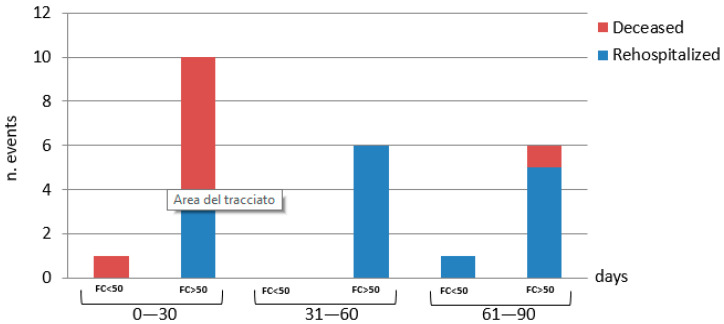
Differences in the combined outcome according to fecal calprotectin values.

**Table 1 medicina-60-00008-t001:** Baseline demographic, clinical, laboratory, and echocardiographic features of total cohort and different subgroups according to fecal calprotectin values.

		Fecal Calprotectin Values		
	Total Cohort (*n*. 59)	<50 µg/g (*n*. 23)	>50 µg/g (*n*. 36)	*p*
**Demographic and clinical features**				
Median age (years)	81 (76–86)	77 (68–85)	84.5 (78.3–87.8)	<0.05
Sex (females, %)	23 (39%)	9 (39.1%)	14 (38.9%)	0.10
NYHA class II (*n*. of pts, %)	9 (15.2%)	5 (21.7%)	4 (11.1%)	0.40
NYHA class III (*n*. of pts, %)	35 (59.3%)	16 (69.6%)	19 (52.8%)	
NYHA class IV (*n*. of pts, %)	15 (20.3%)	6 (26.1%)	9 (25%)	
Ischemic cardiomiopathy (*n*. of pts, %)	21 (35.6%)	9 (39.1%)	21 (58.3%)	0.40
Valvular heart disease (*n*. of pts, %)	12 (20.3%)	7 (30.4%)	5 (13.9%)	
Hypertensive cardiomyopathy (*n*. of pts, %)	5 (8.5%)	2 (8,7%)	3 (8.3%)	
Idiopathic cardiomyopathy (*n*. of pts, %)	4 (6.8%)	1 (1.7%)	3 (8.3%)	
Atrial fibrillation (*n*. of pts, %)	17 (28.8%)	4 (4.3%)	13 (36.1%)	0.38
Diabetes mellitus (*n*. of pts, %)	24 (40.7%)	9 (39.1%)	15 (41.7%)	0.85
ADHF (*n*. of pts, %)	48 (81.4%)	17 (73.9%)	31 (86.1%)	0.28
APO (*n*. of pts, %)	8 (8.5%)	3 (13.0%)	5 (13.9%)	0.08
IRVF (*n*. of pts, %)	2 (3.4%)	1 (4.3%)	1 (2.8%)	0.68
CS (*n*. of pts, %)	1 (1.7%)	0 (0%)	1 (2.8%)	0.42
NT-proBNP (median values) pg/mL	4874 (2752–10,959)	3321 (1678–5657)	6988 (3493–14,965)	<0.001
CRP (median values) mg/L	20.5 (5.1–51.2)	15.7 (2.4–47.4)	22.7 (23.2–64.7)	0.25
IL-6 (median values) ng/L	4.8 (2.7–17)	2.7 (2.7–6.9)	5.6 (3.4–23.8)	0.60
ALT (median values) UI/L	18.5 (12–32)	18 (11–29)	19 (12–42)	0.49
γGT (median values) UI/L	38 (23–64)	38 (21–64)	37.5 (23.2–64.8)	0.99
ALP (median values) UI/L	78.5 (59.3–103.5)	66.5 (52.5–95.3)	85.5 (65–106.5)	0.11
Total serum bilirubin (median values) mg/dL	0.9 (0.6–1.2)	0.7 (0.6–1.2)	0.9 (0.7–1.2)	0.21
Fecal calprotectin (median values) µg/g	117.1 (47–170)	44.5 (11.2–78,7)	157 (122–197)	<0.001
Serum zonulin (median values) ng/mL	54.4 (42.0–78.5)	8.2 (6.9–9.7)	8.4 (7.2–9.3)	0.83
Fecal zonulin (median values) ng/g	8.3 (7.0–9.6)	49.7 (30–77.8)	65 (42.1–82.4)	0.72
TAPSE (mm)	18 (15–20)	19 (17–21)	17 (14–19)	<0.05
PAPS (mmHg)	44 (35–55)	45 (35–55)	40 (35–53)	0.40
EF (median values %)	49 (38.56)	50 (44–56)	48.5 (35–56.5)	0.41
Ascites (*n*. of pts, %)	2 (3.4%)	0 (0%)	2 (5.6%)	0.25
IVC diameter (>21 mm, *n*. of pts, %)	19 (32.2%)	8 (34.8%)	11 (30.6%)	0.52

Abbreviations: pts: patients, NYHA: New York Heart Association, (ADHF): acute decompensated heart failure, OPE: acute pulmonary edema, IRVF: isolated right ventricular failure, CS: cardiogenic shock, NT-proBNP: N-terminal fragment B-type natriuretic peptide, CRP: C-reactive protein, IL-6: Interleukin-6, ALT: alanine aminotransferase, γGT: gamma-glutamyl transferase, ALP: alkaline phosphatase, EF: ejection fraction, TAPSE: tricuspid annular plane excursion, PAPs: systolic pulmonary artery pressure, IVC: inferior vena cava.

**Table 2 medicina-60-00008-t002:** Main HF medications during the acute phase of hospitalization and at the hospital discharge.

		Fecal Calprotectin Values		
**Main HF Treatment Strategies** **during Hospitalization**	**Total Cohort** **(*n.* 59)**	**<50 µg/g** **(*n.* 23)**	**>50 µg/g** **(*n.* 36)**	** *p* **
i.v. loop diuretics (*n*. of patients, %)	57 (96.6%)	21 (91.3%)	35 (97.2%)	0.317
Vasodilators (*n*. of patients, %)	4 (6.8%)	1 (4.3%)	3 (8.3%)	0.556
Inotropes (*n*. of patients, %)	1 (1.7%)	0 (0%)	1 (2.8%)	0.424
Oxygen supplementation (*n*. of patients, %)	48 (81.4%)	18 (78.2%)	30 (83.3%)	0.466
Non-invasive positive pressure ventilation (*n*. of patients, %)	4 (6.8%)	1 (4.3%)	3 (8.3%)	0.076
		**Fecal Calprotectin Values**		
**Main HF Medications at** **Hospital Discharge**	**Total Cohort** **(*n.* 51)**	**<50 µg/g** **(*n.* 22)**	**>50 µg/g** **(*n.* 29)**	** *p* **
Loop diuretics (*n*. of patients, %)	51 (100%)	22 (100%)	29 (100%)	0.327
Beta-blockers (*n*. of patients, %)	51 (100%)	22 (100%)	29 (100%)	0.327
ARNIs (*n*. of patients, %)	33 (64.7%)	10 (45.5%)	23 (79.3%)	0.013
ACE-1s/ARBs (*n*. of patients, %)	46 (90.2%)	19 (86.7%)	27 (93.1%)	0.427
Aldosterone antagonists (*n*. of patients, %)	40 (78.4%)	18(81.8%)	22 (75.9%)	0.071
Ivabradine (*n*. of patients, %)	2 (3.9%)	1 (4.5%)	1 (3.4%)	0.843
SGLT-2 inhibitors (*n*. of patients, %)	26 (51%)	16 (72.7%)	10 (34.5%)	0.007

Abbreviations: AHF: acute heart failure, i.v.: intravenous, ARNI: angiotensin receptor-neprilysin inhibitors, ACE-I: angiotensin-converting enzyme inhibitors, ARBs: angiotensin-II receptor blockers, SGLT-2, sodium-glucose co-transporter 2.

**Table 3 medicina-60-00008-t003:** Univariate analysis of variables correlated to the main combined outcome.

	Rehospitalization and/or Death within 90 Days		
	NO (*n.* pts 40)	YES (*n.* pts 19)	Univariate *p*
Median age (years)	81 (76–80)	82 (73–87)	0.90
Sex (females, %)	13 (32.5%)	10 (52.6%)	0.14
NYHA class IV (*n*. of patients, %)	8 (20%)	11 (57.9%)	<0.05
Ischemic cardiomiopathy (*n*. of patients, %)	14 (35%)	7 (36.8%)	0.86
Valvular heart disease (*n*. of patients, %)	8 (20%)	4 (6.8%)	
Hypertensive cardiomyopathy (*n*. of patients, %)	3 (7.5%)	2 (21.1%)	
Idiopathic cardiomyopathy (*n*. of patients, %)	2 (5.0%)	2 (10.5%)	
Atrial fibrillation (*n*. of patients, %)	13 (32.5%)	4 (21.1%)	0.75
Diabetes mellitus (*n*. of patients, %)	14 (35%)	10 (52.6%)	0.20
NT-proBNP (median values) pg/mL	4550 (1782–6876)	11,917 (3539–21,266)	0.001
CRP (median values) mg/L	17.4 (4.3–47.4)	24.9 (6.8–63.6)	0.24
IL-6 (median values) ng/L	4.5 (2.7–15.4)	6.9 (2.9–26)	0.29
ALT (median values) UI/L	17 (12–24)	24 (14–42)	0.12
γGT (median values) UI/L	37 (21–64.5)	40 (28–64)	0.49
ALP (median values) UI/L	74.5 (57–95.5)	88.5 (69.5–111.5)	0.15
Total serum bilirubin (median values) mg/dl	0.8 (0.6–1.2)	0.9 (0.7–1.2)	0.91
Fecal calprotectin (median values) µg/g	98.5 (44.6–167.7)	121.6 (110.1–176.5)	0.09
Fecal calprotectin (>50 µg/g, *n*. of patients, %)	19 (47.5%)	17 (89.5%)	0.002
Serum zonulin (median values) ng/mL	8.5 (7.4–10.0)	8.3 (6.6–9.2)	0.45
Fecal zonulin (median values) ng/g	49.4 (41.5–79.5)	65 (42.4–76.1)	0.87
TAPSE (mm)	18 (15.9–21)	17 (14.2–19)	0.26
PAPS (mmHg)	40 (35–50)	45 (37.5–58.5)	0.29
EF (median values %)	50 (39.2–56.8)	46 (29–55)	0.33
Ascites (*n*. of patients, %)	0 (0%)	2 (10.5%)	0.06
IVC diameter (>21 mm, *n*. of patients, %)	11 (27.5%)	8 (42.1%)	0.61

NTproBNP value > 4874 pg/mL. Abbreviations: pts: patients, NYHA: New York Heart Association, NT-proBNP: N-terminal fragment B-type natriuretic peptide, CRP: C-reactive protein, IL-6: Interleukin-6, ALT: alanine aminotransferase, γGT: gamma-glutamyl transferase, ALP: alkaline phosphatase, EF: ejection fraction, TAPSE: tricuspid annular plane excursion, PAPs: systolic pulmonary artery pressure, IVC: inferior vena cava.

**Table 4 medicina-60-00008-t004:** Multivariate analysis of variables correlated to the main combined outcome.

	Coefficient Beta	SE	Walds Statistic	Sign	OR	95% CI	
						Inferior	Superior
FC > 50 µg/g	1.823	0.874	4.351	0.037	6.192	1.116	34.351
NT-proBNP > 1249 pg/mL	2.208	0.698	10.013	0.002	9.093	2.317	35.691
Constant	−2.927	0.834	12.318	0.000	0.054		

Abbreviations: FC: fecal calprotectin, NT-proBNP: N-terminal fragment B-type natriuretic peptide; SE: standard error; Sign: significance; OR: Odd Ratios, CI: Confidence interval.

## Data Availability

Data are contained within the article and Appendix A.

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
