# Peer review of "Role of Intestinal Inflammation and Permeability in Patients with Acute Heart Failure"

_medicina, 2023, doi:10.3390/medicina60010008_

Round 1

Reviewer 1 Report

Comments and Suggestions for Authors

The article “Role of intestinal inflammation and permeability in patients with heart failure” is devoted to an interesting and relevant topic, namely the assessment of the prognostic role of intestinal inflammation and permeability in older patients with acute HF, and their correlation with the common parameters traditionally used in the management of acute HF.

This article may be considered for publication after a number of comments have been corrected.

In the introduction and discussion, conclusions, it is necessary to emphasize the study of the role of inflammation in the intestine in patients with acute HF, because the absence in the introduction and discussion of the distinction between the concepts of chronic HF and acute HF is confusing. In my opinion, these two conditions should be separated from each other, because the study group was represented only by patients with acute HF. The introduction needs to be specific.

In the inclusion criteria, it is worth specifying the “minor age” item.

For better clarity, authors may consider including a graphical design of the study in the article file at the discretion of the authors.

It is worth adding the purpose of the study to the main file of the article.

It is recommended to update literature sources; currently, out of 37 sources, 22 are older than 5 years ago.

Author Response

The article “Role of intestinal inflammation and permeability in patients with heart failure” is devoted to an interesting and relevant topic, namely the assessment of the prognostic role of intestinal inflammation and permeability in older patients with acute HF, and their correlation with the common parameters traditionally used in the management of acute HF.

This article may be considered for publication after a number of comments have been corrected.

R: Thank you for your careful revision of our paper and for your precious comments, that improved the quality of our paper. We have done our best to address your comments satisfactorily. Please find below a point-by-point response to your specific comments.

In the introduction and discussion, conclusions, it is necessary to emphasize the study of the role of inflammation in the intestine in patients with acute HF, because the absence in the introduction and discussion of the distinction between the concepts of chronic HF and acute HF is confusing. In my opinion, these two conditions should be separated from each other, because the study group was represented only by patients with acute HF. The introduction needs to be specific.

R: We thank the Reviewer for this precious comment. The text has been amended, better distinguishing the concept of AHF and CHF throughout the manuscript.

In the inclusion criteria, it is worth specifying the “minor age” item.

  1. The text has been modified as requested.

For better clarity, authors may consider including a graphical design of the study in the article file at the discretion of the authors.

  1. We appreciate the Reviewer’s suggestion and added an explicative picture of the design of the study.

It is worth adding the purpose of the study to the main file of the article.

R: The text has been modified as requested.

It is recommended to update literature sources; currently, out of 37 sources, 22 are older than 5 years ago.

R: We agree with the Reviewer’s suggestion and updated the references as indicated

Reviewer 2 Report

Comments and Suggestions for Authors

Title: "Intestinal Inflammation and Permeability in Heart Failure: A Prospective Cohort Study"

Abstract:

The manuscript explores the association between bowel inflammatory markers and prognosis in heart failure (HF) patients through a single-center prospective cohort study. The 59 subjects were observed for 90 days post-discharge. Although the subject matter is intriguing given the limited understanding of the interplay between the intestines and heart failure, several crucial points require attention.

  1. Introduction:

The authors intriguingly propose that both venous congestion and reduced cardiac output in chronic and acute HF induce bowel hypoperfusion, potentially leading to mucosal ischemia. However, the relative harm caused by congestion versus hypoperfusion remains unclear. It is suggested that the authors conduct a detailed analysis, incorporating hemodynamic profiles (e.g., wet-warm), lactates as hypoperfusion indicators, and relevant surrogates of congestion. Alternatively, a comprehensive discussion in the context of existing literature could be provided.

  1. Methods:

A critical flaw arises in the inclusion of patients diagnosed with acute HF solely by emergency department clinicians. It is recommended that the diagnosis align with established guidelines and, at a minimum, be confirmed by a cardiologist.

  1. Participant Exclusion:

A noteworthy 21% loss of participants during follow-up requires elucidation. The rationale behind such a substantial loss should be explicitly addressed to maintain the study's credibility.

  1. Data Analysis:

Table 1 highlights that the group with higher fecal calprotectin exhibited over a two-fold increase in NT-proBNP levels, older age, and worse TAPSE. This prompts the suspicion that these patients, indicative of a worse general status and possibly multi-organ dysfunction, might contribute to poorer outcomes. This hypothesis should be extensively discussed in the Results section.

Considering the ROC curve analysis performed for NT-proBNP, a parallel analysis for calprotectin levels is recommended, offering a more reliable evaluation than a simple reference value.

Clinical characteristics of HF patients, including the predominance of HFpEF, percentage of patients with reduced EF, proportions of chronic decompensation/de novo HF, and hemodynamic profiles, are crucial omissions that should be rectified in the methods and title.

Information on the treatment of patients in both groups is absent, a notable gap that is critical for a comprehensive outcome assessment.

  1. Discussion:

Implications of the results, such as the potential role of calprotectin assessment in predicting responses to oral/intravenous diuretic treatment, should be explored in the discussion.

  1. Typo Correction:

In line 288, it is suggested that there might be a typo in the phrase, "The exact mechanisms underlying the role of calprotectin in HF remain clear."

  1. Conclusions:

The conclusions section requires thorough revision to avoid redundancy and offer a more succinct summary of the key findings without reiterating the introduction.

Comments on the Quality of English Language

Extensive langauge editing required.

Author Response

Title: "Intestinal Inflammation and Permeability in Heart Failure: A Prospective Cohort Study"

Abstract:

The manuscript explores the association between bowel inflammatory markers and prognosis in heart failure (HF) patients through a single-center prospective cohort study. The 59 subjects were observed for 90 days post-discharge. Although the subject matter is intriguing given the limited understanding of the interplay between the intestines and heart failure, several crucial points require attention.

R: Thank you for your careful revision of our paper and for your precious comments, that improved the quality of our paper. We have done our best to address your comments satisfactorily. Please find below a point-by-point response to your specific comments.

  1. Introduction:

The authors intriguingly propose that both venous congestion and reduced cardiac output in chronic and acute HF induce bowel hypoperfusion, potentially leading to mucosal ischemia. However, the relative harm caused by congestion versus hypoperfusion remains unclear. It is suggested that the authors conduct a detailed analysis, incorporating hemodynamic profiles (e.g., wet-warm), lactates as hypoperfusion indicators, and relevant surrogates of congestion. Alternatively, a comprehensive discussion in the context of existing literature could be provided.

  1. We thank the Reviewer for this precious comment. In Introduction, we add a paragraph about the different hemodynamic profiles of HF, mainly focusing on current guidelines.

Methods:

A critical flaw arises in the inclusion of patients diagnosed with acute HF solely by emergency department clinicians. It is recommended that the diagnosis align with established guidelines and, at a minimum, be confirmed by a cardiologist.

  1. We thank the Reviewer for this comment. We better specified in the Method Section that diagnosis of HF was made according to Clinicians’ judgement, however always based on current guidelines. All the enrolled patients, in fact, fitted with the current definition of AHF (ESC 2021). In any case, the ED of our tertiary center is supplied with a Cardiologist Consultant, who supports the other Physicians in diagnosis and management of such patients. Also this element was added in the test.

A noteworthy 21% loss of participants during follow-up requires elucidation. The rationale behind such a substantial loss should be explicitly addressed to maintain the study's credibility.

  1. We thank the Reviewer and agree with his concern. We apologize for the incorrect sentence “sixteen subjects were excluded from the final analysis since they were lost at follow-up”. Actually, the most patients (11 patients) excluded from the final analysis, consisted in those subjects giving their informed consent at time of admission in the ED, but unable to provide stool sample within the required 48 hours. Since the main outcome of the study was to evaluate the prognostic role of fecal calprotectin, we did not include these subjects in the final analysis. Instead, five patients did not complete the total 90-days follow-up by phone interview. We better specified these data in the Result Section.  
  2. Data Analysis

Table 1 highlights that the group with higher fecal calprotectin exhibited over a two-fold increase in NT-proBNP levels, older age, and worse TAPSE. This prompts the suspicion that these patients, indicative of a worse general status and possibly multi-organ dysfunction, might contribute to poorer outcomes. This hypothesis should be extensively discussed in the Results section.

  1. We thank the Reviewer for this thoughtful question. Actually, we excluded from the logistic models the variables showing high collinearity, both to reduce overfitting and to improve model stability. TAPSE and NYHA showed high collinearity with NT-proBNP values, and for this reason, we preferred to use just the latter in the models. This is now better specified in the text in the statistical methods section.

Considering the ROC curve analysis performed for NT-proBNP, a parallel analysis for calprotectin levels is recommended, offering a more reliable evaluation than a simple reference value.

  1. We thank the Reviewer for this suggestion. We discussed this point before the paper submission and finally, we decided to use the standard cut-off for calprotectin. This choice is due to making our results more generalizable, compared to a cut-off point derived from the ROC analysis that indeed would have been dependent on the study cohort used. Nevertheless, we used the Youden index J cut-off for NT-proBNP since all the patients in the cohort had a baseline value above the standard reference value.

We agree with the Reviewer that this point would have needed a more detailed insight. For this reason, we added supplementary data with the ROC curve analysis for NT-proBNP and Calprotectin and a table with a separate logistic regression model obtained by dividing calprotectin values by Youden cut-off. Since the results were pretty similar to the standard cut-off we did not include this latter logistic model in the main text.

We also had the chance to correct the cut-off value for NT-proBNP in the table that was erroneously reported as 1249, instead of 4874 for a writing typo.

Clinical characteristics of HF patients, including the predominance of HFpEF, percentage of patients with reduced EF, proportions of chronic decompensation/de novo HF, and hemodynamic profiles, are crucial omissions that should be rectified in the methods and title.

  1. We thank the Reviewer. We added in the Result Section a paragraph including the number of patients with preserved or reduced EF, and the proportion of decompensation/de novo HF. As reported in Introduction, the classical distinction between “wet-warm” haemodynamic profiles was partially replaced by current guidelines (ESC 2021). By reviewing medical records, therefore, we categorized our cohort according to the main clinical presentation (i.e. acute decompensated heart failure (ADHF), acute pulmonary edema (APE), isolated right ventricular failure (IRVF), cardiogenic shock (CS). We also updated the Method Section.  

Information on the treatment of patients in both groups is absent, a notable gap that is critical for a comprehensive outcome assessment.

  1. We thank the Reviewer. As specified in the previous comment, the most predominant clinical presentation in our cohort, was systemic congestion. Therefore, the mainstay of treatment was represented by iv diuretics and oxygen supplementation, without significant differences among the two groups. However, we added complete information regarding patient treatment in the Result section.
  2. Discussion:

Implications of the results, such as the potential role of calprotectin assessment in predicting responses to oral/intravenous diuretic treatment, should be explored in the discussion.

We thank the Reviewer for this precious comment and we added a paragraph in Discussion about this topic.

  1. Typo Correction:

In line 288, it is suggested that there might be a typo in the phrase, "The exact mechanisms underlying the role of calprotectin in HF remain clear."

We corrected this mistake

  1. Conclusions:

The conclusions section requires thorough revision to avoid redundancy and offer a more succinct summary of the key findings without reiterating the introduction.

We agree with the Reviewer and modified the Conclusion section accordingly

Extensive language editing required

We agree with the Reviewer and revised the text

Round 2

Reviewer 2 Report

Comments and Suggestions for Authors

1. We thank the Reviewer for this precious comment. In Introduction, we add a paragraph about the different hemodynamic profiles of HF, mainly focusing on current guidelines.

I meant the authors should add a paragraph about the implications of different hemodynamic profiles on the intestinal inflamation.I would rather put that on the discussion section, not the introduction.

2. We thank the Reviewer. We added in the Result Section a paragraph including the number of patients with preserved or reduced EF, and the proportion of decompensation/de novo HF. As reported in Introduction, the classical distinction between “wet-warm” haemodynamic profiles was partially replaced by current guidelines (ESC 2021). By reviewing medical records, therefore, we categorized our cohort according to the main clinical presentation (i.e. acute decompensated heart failure (ADHF), acute pulmonary edema (APE), isolated right ventricular failure (IRVF), cardiogenic shock (CS). We also updated the Method Section.  

It should be added to Table 1 as well.

3. We thank the Reviewer. As specified in the previous comment, the most predominant clinical presentation in our cohort, was systemic congestion. Therefore, the mainstay of treatment was represented by iv diuretics and oxygen supplementation, without significant differences among the two groups. However, we added complete information regarding patient treatment in the Result section.

It should be added to the Table, and I meant the post-discharge treatment.

The rest of the remarks have been succesfully addressed.

Author Response

  1. We thank the Reviewer for this precious comment. In Introduction, we add a paragraph about the different hemodynamic profiles of HF, mainly focusing on current guidelines.

 I meant the authors should add a paragraph about the implications of different hemodynamic profiles on the intestinal inflamation.I would rather put that on the discussion section, not the introduction.

  1. We moved this paragraph to the Discussion section
  2. We thank the Reviewer. We added in the Result Section a paragraph including the number of patients with preserved or reduced EF, and the proportion of decompensation/de novo HF. As reported in Introduction, the classical distinction between “wet-warm” haemodynamic profiles was partially replaced by current guidelines (ESC 2021). By reviewing medical records, therefore, we categorized our cohort according to the main clinical presentation (i.e. acute decompensated heart failure (ADHF), acute pulmonary edema (APE), isolated right ventricular failure (IRVF), cardiogenic shock (CS). We also updated the Method Section.  

 It should be added to Table 1 as well.

  1. We added these data in Table 1.
  2. We thank the Reviewer. As specified in the previous comment, the most predominant clinical presentation in our cohort, was systemic congestion. Therefore, the mainstay of treatment was represented by iv diuretics and oxygen supplementation, without significant differences among the two groups. However, we added complete information regarding patient treatment in the Result section.

 It should be added to the Table, and I meant the post-discharge treatment.

  1. We added a new Table (Table 2) incorporating data about the main AHF therapies performed during hospitalization and the HF medications at discharge.